

# Enhanced related-key differential neural distinguishers for SIMON and SIMECK block ciphers

Gao Wang[1] and Gaoli Wang[1,2]

[1] Shanghai Key Laboratory of Trustworthy Computing, Software Engineering Institute, East China Normal University, Shanghai, North Zhongshan Road, China

[2] Advanced Cryptography and System Security Key Laboratory of Sichuan Province, Sichuan Province, Chengdu, China

## ABSTRACT

At CRYPTO 2019, Gohr pioneered the application of deep learning to differential cryptanalysis and successfully attacked the 11-round NSA block cipher Speck32/64 with a 7-round and an 8-round single-key differential neural distinguisher. Subsequently, Lu et al. (DOI 10.1093/comjnl/bxac195) presented the improved related-key differential neural distinguishers against the SIMON and SIMECK. Following this work, we provide a framework to construct the enhanced related-key differential neural distinguisher for SIMON and SIMECK. In order to select input differences efficiently, we introduce a method that leverages weighted bias scores to approximate the suitability of various input differences. Building on the principles of the basic related-key differential neural distinguisher, we further propose an improved scheme to construct the enhanced related-key differential neural distinguisher by utilizing two input differences, and obtain superior accuracy than Lu et al. for both SIMON and SIMECK. Specifically, our meticulous selection of input differences yields significant accuracy improvements of 3% and 1.9% for the 12-round and 13-round basic related-key differential neural distinguishers of SIMON32/64. Moreover, our enhanced related-key differential neural distinguishers surpass the basic related-key differential neural distinguishers. For 13-round SIMON32/64, 13-round SIMON48/96, and 14-round SIMON64/128, the accuracy of their related-key differential neural distinguishers increases from 0.545, 0.650, and 0.580 to 0.567, 0.696, and 0.618, respectively. For 15-round SIMECK32/64, 19-round SIMECK48/96, and 22-round SIMECK64/128, the accuracy of their neural distinguishers is improved from 0.547, 0.516, and 0.519 to 0.568, 0.523, and 0.526, respectively.

# INTRODUCTION

In recent years, with the wide application of wireless sensor networks (WSN) and radio frequency identification (RFID) technology in various industries, the data security problem of these resource-constrained devices have become more and more prominent. As a cryptographic solution that can achieve a good balance between security and performance under limited resources, lightweight block ciphers are widely used to protect data security in various resource-constrained devices. The security of block ciphers is closely related to the security of data. In this context, evaluating the security properties of

Corresponding author
Gaoli Wang, glwang@sei.ecnu.edu.cn

these ciphers has become a popular research topic in the field of computer science and cryptography. Among many cryptanalysis techniques, differential cryptanalysis, proposed by *Biham & Shamir (1991b)*, is one of the most commonly used methods for evaluating the security of block ciphers. This technique focuses on the propagation of plaintext differences during the encryption.

In traditional differential cryptanalysis, the core task of differential cryptanalysis is to find a differential characteristic with high probability. Initially, this task was achieved by manual derivation, which required a lot of effort and time. At EUROCRYPT 1994, *Matsui (1994)* presented a branch-and-bound method for this task, which replaced manual derivation with automated search techniques for the first time. However, for the block ciphers with large sizes, this method is insufficient to provide useful differential characteristics. This prompts cryptographers to adopt more efficient automated search tools for searching the differential characteristic with high probability, including mixed integer linear programming (MILP) (*Sun et al., 2014*; *Bellini et al., 2023a*; *Mouha et al., 2012*), constraint programming (CP) (*Gerault, Minier & Solnon, 2016*; *Sun et al., 2017*), and Boolean satisfiability problem or satisfiability modulo theories (SAT/SMT) (*Sun et al., 2017*; *Lafitte, 2018*).

In recent years, with the rapid development of deep learning, cryptanalysts have begun to explore how to harness its power for differential cryptanalysis. At CRYPTO 2019, *Gohr (2019)* constructed an 8-round differential neural distinguishers by leveraging neural networks to learn the differential properties of block ciphers SPECK32/64 and successfully carried out an 11-round key recovery attack. This pioneering research significantly accelerated the integration of deep learning and differential cryptanalysis. Since this study, the differential neural distinguisher has been widely applied to various block ciphers in single-key and related-key scenarios, including but not limited to SIMON (*Bao et al., 2022*; *Lu et al., 2024*; *Bellini et al., 2023b*), SIMECK (*Zhang et al., 2023*; *Lu et al., 2024*), PRESENT (*Jain, Kohli & Mishra, 2020*; *Bellini et al., 2023b*; *Zhang, Wang & Chen, 2023*), GIFT (*Shen et al., 2024*), ASCON (*Shen et al., 2024*), and others. In most of these works, the focus is only on the single-key neural distinguishers, while SIMON and SIMECK also focus on the related-key differential neural distinguishers. In this article, we continue to optimize the related-key differential neural distinguishers for SIMON and SIMECK.

So far, there are many studies exploring the differential neural distinguishers for SIMON and SIMECK ciphers, such as *Bao et al. (2022)*, *Zhang et al. (2023)*, *Wang et al. (2022)*, *Seong et al. (2022)*, *Gohr, Leander & Neumann (2022)*, *Lyu, Tu & Zhang (2022)*, *Lu et al. (2024)*. However, most of them focused on the single-key scenario, until the research of *Lu et al. (2024)* broke this trend. They not only improved the accuracy of their single-key differential neural distinguishers by using the enhanced data format $(\Delta_L^r, \Delta_R^r, C_l, C_r, C_l', C_r', \Delta_R^{r-1}, p\Delta_R^{r-2})$ (defined in Eq. (10)), but also constructed the related-key differential neural distinguishers for them. The experimental results show that the related-key differential neural distinguishers outperforms the single-key differential neural distinguishers in terms of the number of analyzed rounds and accuracy. In the single-key scenario, *Lu et al. (2024)* exhaustively evaluated the input differences with Hamming weights of 1, 2, and 3 by training a differential neural distinguisher for each difference. However, for the related-key

scenario, this task has not been explored in depth due to the huge number of input differences that need to be evaluated. Even for the smallest variants SIMON32/64 and SIMECK32/64, the number of input differences with Hamming weights of 1, 2, and 3 already reaches about 200 million. Therefore, it is impractical to train a neural distinguisher for each difference. In this article, we aim to further address this challenge.

## Our contributions

In this article, we first present a framework to construct the basic related-key differential neural distinguishers for SIMON and SIMECK. This framework is comprised of five components: differences selection, sample generation, network architecture, distinguisher training, and distinguisher evaluation. For comparison with the baseline work of *Lu et al. (2024)*, we keep sample generation, network architecture, distinguisher training, and distinguisher evaluation as in *Lu et al. (2024)*. Our attention is mainly on differences selection. We provide a method for approximately assessing the suitability of different input differences with weighted bias scores instead of training a neural distinguisher for every input difference as *Lu et al. (2024)*. This allows us to approximate the applicability of the different differences without training the model, which can significantly accelerates the process of differences selection. Our meticulous selection of the input difference can make the accuracy of the basic related-key differential neural distinguisher match or surpass previous results. In particular, the accuracy for the 12-round and 13-round distinguishers of SIMON32/64 is improved from 0.648 and 0.526 to 0.678 and 0.545, respectively, as shown in Table 1.

Furthermore, based on the principles of the basic related-key differential neural distinguishers, we propose an enhanced scheme that harnesses two distinct input differences to construct a more powerful related-key differential neural distinguisher for SIMON and SIMECK instead of using only one difference in the phase of sample generation.

Specifically, for the 13-round SIMON32/64, 13-round SIMON48/96, and 14-round SIMON64/128, their accuracy is raised from 0.545, 0.650, and 0.580 to 0.567, 0.696, and 0.618, respectively. Similarly, the neural distinguishers for 15-round SIMECK32/64, 19-round SIMECK48/96, and 22-round SIMECK64/128 also showed significant improvements in accuracy, rising from 0.547, 0.516, and 0.519 to 0.568, 0.523, and 0.526, respectively. All these results illustrate the effectiveness and robustness of our scheme.

## Organization

"Preliminaries" commences by introducing the foundational knowledge about the related-key differential neural distinguisher. Following this, "The Framework for Developing Related-Key Differential Neural Distinguishers to SIMON and SIMECK" comprehensively explores the construction of basic and enhanced neural distinguishers for SIMON and SIMECK. Building upon this framework, "Related-Key Differential Neural Distinguishers for Round Reduced SIMON and SIMECK" constructs the improved related-key differential neural distinguishers for SIMON and SIMECK. Finally, "Conclusions and Future Work" concludes this article.

**Table 1 Summary of related-key neural distinguishers against SIMON32/64, SIMON48/96, SIMON64/128, SIMECK32/64, SIMECK48/96, and SIMECK64/128 using eight pairs of ciphertexts as a sample.**

| Cipher | Round | Model | Acc | TPR | TNR | Source |
|---|---|---|---|---|---|---|
| SIMON32/64 | 12 | RKND | 0.648 | 0.652 | 0.644 | *Lu et al. (2024)* |
| | | RKND | 0.678 | 0.685 | 0.671 | "Basic Related-Key Differential Neural Distinguishers" |
| | | RKND' | 0.740 | 0.729 | 0.750 | "Enhanced Related-Key Differential Neural Distinguishers" |
| | 13 | RKND | 0.526 | 0.544 | 0.508 | *Lu et al. (2024)* |
| | | RKND | 0.545 | 0.537 | 0.552 | "Basic Related-Key Differential Neural Distinguishers" |
| | | RKND' | 0.567 | 0.564 | 0.570 | "Enhanced Related-Key Differential Neural Distinguishers" |
| SIMON48/96 | 12 | RKND | 0.993 | 0.999 | 0.986 | "Basic Related-Key Differential Neural Distinguishers" |
| | | RKND' | 0.997 | 0.998 | 0.996 | "Enhanced Related-Key Differential Neural Distinguishers" |
| | 13 | RKND | 0.650 | 0.660 | 0.640 | "Basic Related-Key Differential Neural Distinguishers" |
| | | RKND' | 0.696 | 0.698 | 0.695 | "Enhanced Related-Key Differential Neural Distinguishers" |
| SIMON64/128 | 13 | RKND | 0.840 | 0.839 | 0.841 | *Lu et al. (2024)* |
| | | RKND' | 0.916 | 0.910 | 0.922 | "Enhanced Related-Key Differential Neural Distinguishers" |
| | 14 | RKND | 0.579 | 0.589 | 0.568 | *Lu et al. (2024)* |
| | | RKND' | 0.618 | 0.596 | 0.639 | "Enhanced Related-Key Differential Neural Distinguishers" |
| SIMECK32/64 | 14 | RKND | 0.668 | 0.643 | 0.693 | *Lu et al. (2024)* |
| | | RKND' | 0.730 | 0.722 | 0.738 | "Enhanced Related-Key Differential Neural Distinguishers" |
| | 15 | RKND | 0.547 | 0.517 | 0.576 | *Lu et al. (2024)* |
| | | RKND' | 0.568 | 0.553 | 0.582 | "Enhanced Related-Key Differential Neural Distinguishers" |
| SIMECK48/96 | 18 | RKND | 0.551 | 0.456 | 0.646 | "Basic Related-Key Differential Neural Distinguishers" |
| | | RKND' | 0.572 | 0.572 | 0.572 | "Enhanced Related-Key Differential Neural Distinguishers" |
| | 19 | RKND | 0.516 | 0.411 | 0.611 | "Basic Related-Key Differential Neural Distinguishers" |
| | | RKND' | 0.523 | 0.527 | 0.518 | "Enhanced Related-Key Differential Neural Distinguishers" |
| SIMECK64/128 | 21 | RKND | 0.552 | 0.425 | 0.679 | *Lu et al. (2024)* |
| | | RKND' | 0.572 | 0.580 | 0.563 | "Enhanced Related-Key Differential Neural Distinguishers" |
| | 22 | RKND | 0.518 | 0.391 | 0.646 | *Lu et al. (2024)* |
| | | RKND' | 0.526 | 0.523 | 0.529 | "Enhanced Related-Key Differential Neural Distinguishers" |

**Note:**
Acc, Accuracy; TPR, True positive rate; TNR, True negative rate. *RKND*: The basic related-key differential neural distinguisher trained with a difference. *RKND'*: The enhanced related-key differential neural distinguisher trained using a pair of differences.

## PRELIMINARIES

In this section, we first present the pivotal notations in Table 2. Following this, we offer a succinct overview of the block ciphers SIMON and SIMECK, along with the basic concepts about related-key differential cryptanalysis and convolutional neural networks.

### Notations

Table 2 illustrates the notations utilized in this article.

### A brief description of SIMON and SIMECK ciphers

SIMON (*Beaulieu et al., 2015*) is a lightweight block cipher, designed by the National Security Agency (NSA) in 2013. It employs a Feistel structure, making it suitable for resource-constrained environments. In addition, it supports various block lengths and key

**Table 2 Notations.**

| Notation | Description |
|---|---|
| $\lll, \ggg$ | Circular left and right shift |
| $\alpha, \beta, \gamma$ | Bits of cyclic shift |
| $C, Z_j$ | Predefined constants |
| $T$ | Temporary variable |
| $\oplus, \odot$ | Bit-wise XOR and AND operation |
| $\|$ | Concatenation |
| $\mathbb{F}_2$ | Binary field |
| $P, P'$ | Plaintext |
| $C, C'$ | Ciphertext |
| $K, K'$ | Master key |
| $E, R$ | Encryption algorithm and rounds |
| $P_i, K_i, C_i$ | Plaintext, key and ciphertext for round $i$ |
| $t_2, t_1, t_0, k_0$ | Components of the $K$ |
| $\Delta P$ | Plaintext difference |
| $\Delta C$ | Ciphertext difference |
| $\Delta K$ | Master key difference |
| $\Delta P_r$ | The $r$-round input difference |
| $\Delta C_r$ | The $r$-round ciphertext difference |
| $\Delta K_r$ | The $r$-round key difference |
| $\sigma$ | Activation function |
| $x_i$ | Input of the $i$-th neuron |
| $w_i$ | Weight of the $x_i$ |
| $b$ | Bias of the neuron |
| $\mathbf{F}_{tr}$ | The transformation operation of SENet |
| $\mathbf{F}_{sq}(\cdot)$ | The squeeze operation of SENet |
| $\mathbf{F}_{ex}(\cdot, \mathbf{W})$ | The excitation operation of SENet |
| $\mathbf{F}_{scale}(\cdot, \cdot)$ | The channel-wise multiplication of SENet |
| $b_r$ | Bias scores for $r$ rounds |
| $\tilde{b}_r^t$ | Approximate bias score calculated using $t$ samples for $r$ rounds |
| $S_R$ | $R$-rounds weighted bias score |
| $\Omega_i$ | $i$-th ($1 \le i \le 8$) ciphertext pair |
| $l_i$ | Cyclic learning rate for the $i$-th epoch |
| $a, b, n$ | The parameter for calculating $l_i$, defaulting to 0.0001, 0.003 and 29, respectively |
| $c$ | The parameter for L2 regularization, default 0.00001 |
| $bl, kl$ | block length and key length |

sizes, such as `SIMON32/64`, `SIMON48/96`, and `SIMON64/128`, where the first number represents the block length and the second number denotes the key size. The round function of `SIMON` is composed of three simple operations: bit-wise XOR $\oplus$, bit-wise AND $\odot$, and circular left shift $\lll$ operations, as shown in Fig. 1. The round function can be formally defined as:

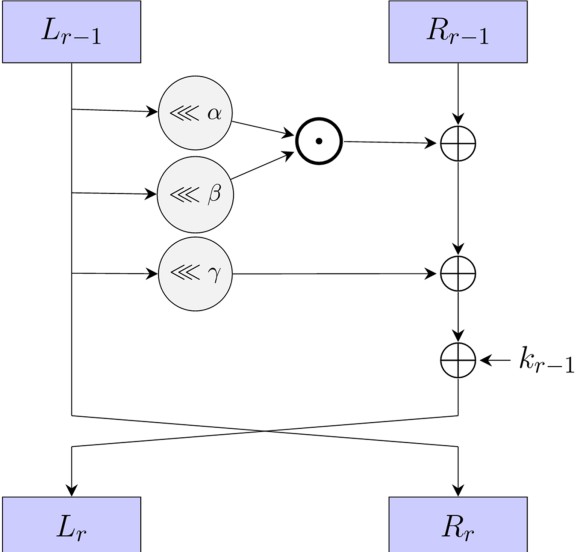

**Figure 1 The round function of SIMON and SIMECK.**

$$\begin{cases} L_r = ((L_{r-1} \lll \alpha) \odot (L_{r-1} \lll \beta)) \oplus R_{r-1} \oplus (L_{r-1} \lll \gamma) \oplus k_{r-1}, \\ R_r = L_{r-1}, \end{cases} \tag{1}$$

where $\alpha$, $\beta$ and $\gamma$ represent the fixed rotation constants that are utilized in the circular left shift operation. For SIMON, the values of these constants are set to 1, 8, and 2, respectively. Given a master key $K$ that comprises four key words, denoted as $K = (K_3, \dots, K_1, K_0)$, the round key $K_{r-1}$ is generated through a linear key schedule. This process incorporates predefined constants $C$ and a series of constants $(Z_j)_i$, the generation follows the scheme outlined below:

$$\begin{cases} T = (K_{i+3} \ggg 3) \oplus K_{i+1}, \\ K_{i+4} = C \oplus (Z_j)_i \oplus K_i \oplus T \oplus (T \ggg 1). \end{cases} \tag{2}$$

SIMON is designed to be highly efficient in terms of both hardware and software implementations. It processes the plaintext and ciphertext blocks in a symmetric manner. Its structure and the choice of operations contribute to its resistance against common attacks like differential cryptanalysis and linear cryptanalysis. Like many lightweight block ciphers, SIMON's simplicity may make it more susceptible to side-channel attacks, such as power analysis and timing attacks.

The SIMECK (*Yang et al., 2015*) cipher, presented at CHES in 2015, is a variant of the SIMON. It retains the same Feistel structure and round function as SIMON, but distinguishes itself through the values of $\alpha$, $\beta$, and $\gamma$, which are set to 0, 5, and 1, respectively. In addition, SIMECK uses the round function to generate the round keys $K_r$ for a given master key $K = (t_2, t_1, t_0, k_0)$, as explained below:

$$\begin{cases} k_{i+1} = t_i, \\ t_{i+3} = k_i \oplus t_i \odot (t_i \lll 5) \oplus (t_i \lll 1) \oplus C \oplus (Z_j)_i. \end{cases} \tag{3}$$

where $C$ and $(Z_j)_i$ are the predefined constants. For more details, please refer to *Yang et al. (2015)*.

SIMECK is a lightweight block cipher designed specifically for constrained environments. It boasts compact hardware implementations and low power consumption, making it suitable for embedded systems and IoT devices. SIMECK has fixed block size and key length, which facilitate consistent and predictable performance. SIMECK stands out for its efficiency in terms of both area and energy, as well as its resistance to common cryptographic attacks. However, it is worth noting that careful implementation is crucial to mitigate potential side-channel attack.

## Related-key differential cryptanalysis

In 1990, *Biham & Shamir (1991b)* introduced a groundbreaking attack strategy called differential cryptanalysis. This cryptanalysis technique can distinguish the block cipher from the random permutation by studying the propagation properties of the plaintext difference $\Delta P$ throughout the encryption. Due to its simple principle and excellent efficacy, this approach quickly attracted significant attention among the cryptography community (*Biham & Shamir, 1991a*, *1992*; *Biham & Dunkelman, 2007*).

In lightweight block ciphers, the key schedule holds paramount importance, as it is responsible for generating and updating the round keys. To delve into the security of this vital component, *Biham (1994)* proposed a pioneering related-key cryptanalysis method in 1994, which studies the security of block cipher under different keys. The related-key differential cryptanalysis method combines the principles of differential cryptanalysis and related-key cryptanalysis. It investigates differential propagation under different keys instead of the same key. The basic concepts related to block cipher and related-key differential cryptanalysis are summarized as follows.

Assuming $E$ is the $r$-round encryption procedure employed by a block cipher with the block length $bl$ and the key length $kl$, and the plaintext, ciphertext, and master key are denoted as $P$, $C$, and $K$, respectively. The formalized encryption process of this block cipher can be expressed as $C = E_K(P)$, which indicates that the ciphertext $C$ results from encrypting the plaintext $P$ for $r$ rounds using the master key $K$. For iterative block ciphers, their encryption process $E_K(P)$ is derived by repeatedly applying the round function $F(K_i, P_i)$, where $K_i$ represents the round key for the $i$-th iteration, whereas $P_i$ denotes the input to this iteration. Consequently, the encryption process of iterative block cipher is given in Eq. (4).

$$E_K(P) = F_{K_r}(P_r) \cdot F_{K_{r-1}}(P_{r-1}) \cdot \ldots \cdot F_{K_2}(P_2) \cdot F_{K_1}(P_1). \tag{4}$$

**Definition 1 (Plaintext Difference, Ciphertext Difference, and Key Difference.** *Matsui (1994)*) *For a block cipher, the plaintext difference $\Delta P$ of the plaintext pair $(P, P')$ is $P \oplus P'$. Similarly, the ciphertext difference $\Delta C$ of the ciphertext pair $(C, C')$ is $C \oplus C'$, and the key difference $\Delta K$ of the key pair $(K, K')$ is $K \oplus K'$.*

**Definition 2 (Related-key Differential Characteristic.** *Jakimoski & Desmedt (2003)*) *Given a plaintext pair $(P, P')$ and a key pair $(K, K')$ with the difference of $\Delta P$ and $\Delta K$, let*

$(C_i, C_i')$ be the cipher pair obtained by encrypting the $(P, P')$ with $(K, K')$ for i rounds, the r-round related-key differential characteristic of the block cipher is $(\Delta P, \Delta C_1, \ldots\ldots, \Delta C_{r-1}, \Delta C_r)$, where $\Delta C_i = C_i \oplus C_i'$.

**Definition 3 (Related-key Differential Probability.** *Jakimoski & Desmedt (2003)*) *The related-key differential probability* $DP(\Delta P, \Delta K, \Delta C)$ *of the block cipher with the plaintext difference* $\Delta P$, *master key difference* $\Delta K$, *and ciphertext difference* $\Delta C$ *is*

$$DP(\Delta P, \Delta K, \Delta C) = \frac{\#\{E_{k\oplus\Delta K}(x \oplus \Delta P) \oplus E_k(x) = \Delta C\}}{2^{|P|+|K|}}, \tag{5}$$

*where* $x \in \mathbb{F}_2^{|P|}$ *and* $k \in \mathbb{F}_2^{|K|}$.

**Definition 4 (Hamming Weight.** *Wang, Wang & He (2021)*) *Assuming* $X \in \mathbb{F}_2^n$, *the hamming weight of is the number of non-zero bits within its binary representation. Mathematically, it can be formulated as* $\sum_{i=1}^n X_i$, *where* $X_i$ *denotes the i-th bit in the binary of X.*

## Convolutional neural network

Convolutional Neural Network (CNN), as a feed-forward neural network with convolutional structure, has been widely applied in numerous domains, including but not limited to image recognition (*Chauhan, Ghanshala & Joshi, 2018*), video analysis (*Ullah et al., 2017*), and natural language processing (*Yin et al., 2017*), and among others. A convolutional neural network usually consists of the input layer, convolutional layer, pooling layer, fully connected layer, and output layer. The convolutional layer is used to extract features, the pooling layer is used to achieve data dimensionality reduction through subsampling, the fully connected layer integrates the previously extracted features for tasks such as classification or regression, and the output layer is responsible for producing the final results.

LeNet-5 is a convolutional neural network designed by *LeCun et al. (1998)* for handwritten digit recognition, and it is one of the most representative results of the early convolutional neural network. It consists of one input layer, one output layer, two convolutional layers, two pooling layers, and two fully connected layers, as shown in Fig. 2. Its input is a image of $32 \times 32$. After two convolution and subsampling operations, this input becomes a feature map of $16 \times 5 \times 5$. The convolution kernels are all $5 \times 5$ with stride 1. The subsampling function used for the pooling layers is maxpooling. Then it passes through two fully connected layers with sizes of 120 and 64 to reach the output layer.

Later, based on LeNet-5, many improved convolutional neural networks have been proposed, such as AlexNet (*Krizhevsky, Sutskever & Hinton, 2017*), GoogleLeNet (*Szegedy et al., 2015*), ResNet (*He et al., 2016*), and so on. The main components used in this article are convolutional layers, activation functions, fully connected layers, as well as the advanced architectures including the Residual Network (ResNet) (*He et al., 2016*) and the Squeeze-and-Excitation Network (SENet) (*Hu, Shen & Sun, 2018*).

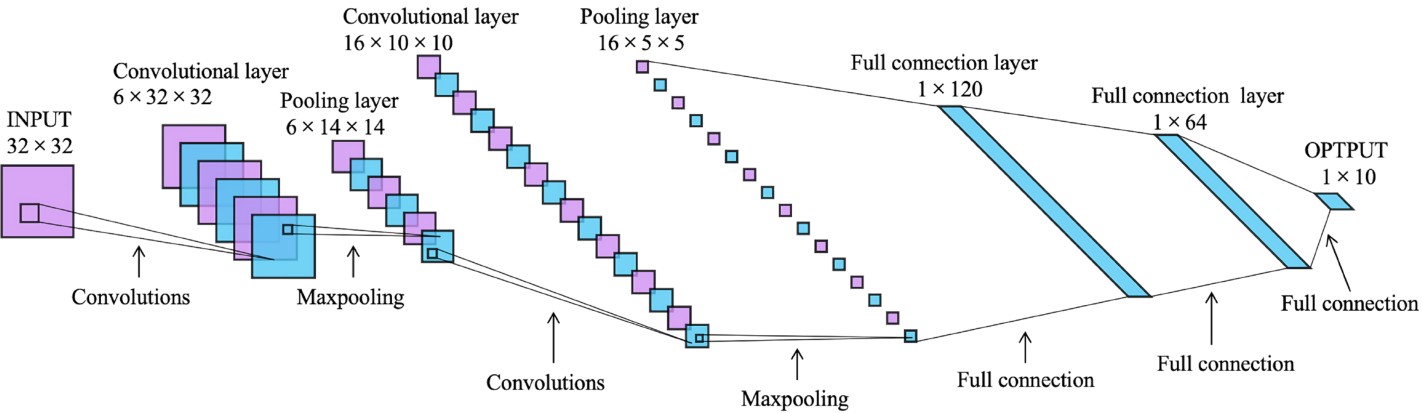

**Figure 2** The architecture of LeNet-5 (*LeCun et al., 1998*).

### Convolution layer

Convolutional layers are the core component of convolutional neural networks. It is responsible for extracting features from input data through convolution operations. In a convolution operations, a convolutional kernel (also known as a filter) continuously slides over the input feature map. At each step, it calculates the sum of the product of the values at each position and takes it as the value in the corresponding position on the output feature map.

### Activation function

In neural networks and deep learning, the activation function plays a crucial role in introducing nonlinear properties that enable the neural network to learn complex patterns in the data. The activation functions Sigmoid (*Little, 1974*) and rectified linear unit (ReLU) (*Nair & Hinton, 2010*) are used in this article. The Sigmoid function can map any real value to an output between 0 and 1. Therefore, it is a common choice for the output layer in binary classification problems. The ReLU function returns the input value itself for the positive inputs and zero for the negative inputs. It performs well in many deep learning tasks because of its effectiveness in mitigating the gradient vanishing problem. Their mathematical formulations are as follows:

$$\textbf{Sigmoid}: f(x) = \frac{1}{1 + e^{-x}}, \quad \textbf{ReLU}: f(x) = max(0, x). \tag{6}$$

### Fully connected layer

The fully connected layer (also known as dense layer) is a fundamental element of neural networks. In this layer, every neuron establishes a connection to each neuron in the preceding layer. This connection ensures that all the outputs from the previous layer are the inputs to every neuron in the current layer. This structure allows the fully connected

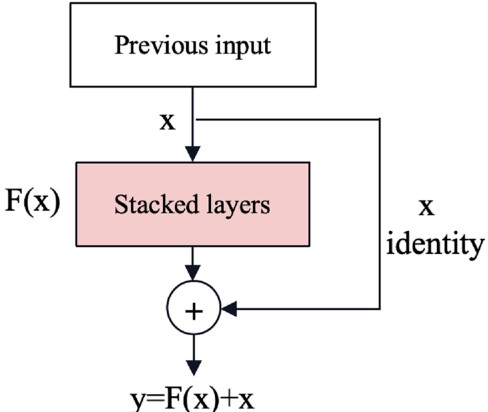

**Figure 3 The shortcut connections of ResNet (*He et al., 2016*).**

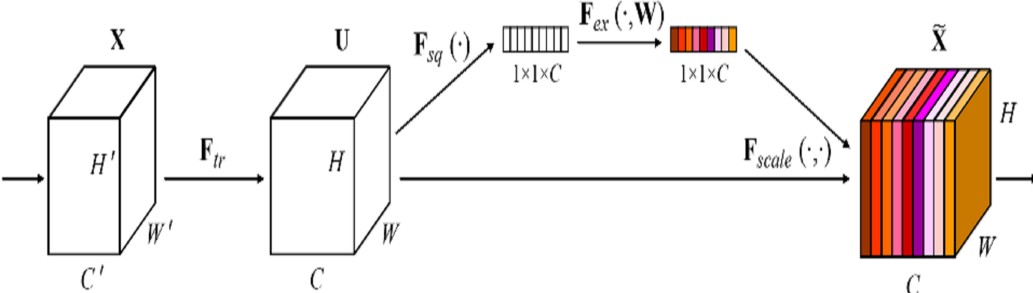

**Figure 4 The squeeze-and-excitation block of SENet (*Hu, Shen & Sun, 2018*).**

layer to execute a weighted combination of input features, effectively capturing the intricate relationships between them. For a single neuron in the fully connected layer, its output can be represented as $\sigma\left(\sum_{i=1}^{n} w_i \cdot x_i + b\right)$, where $n$ is the total number of neurons, $\sigma$ represents the activation function, $x_i$ denotes the input of the $i$-th neuron, $w_i$ corresponds to the weight of the connection, and $b$ is the bias of the neuron.

### Residual network

Gradient vanishing and explosion are issues in deep neural networks where gradients become extremely small or large during back-propagation, respectively, hindering effective training. The residual neural network (ResNet) (*He et al., 2016*) is an effective deep learning model that solves the problem of gradient vanishing and gradient explosion by introducing shortcut connections as shown in Fig. 3. Shortcut connections can mitigate these problems by providing alternative paths for gradient flow, reducing the dependency on gradients passing through all intermediate layers and improving information flow through the network.

### Squeeze-and-excitation network (SENet)

The Squeeze-and-Excitation (SE) block (*Hu, Shen & Sun, 2018*) is a plug-and-play channel attention mechanism that can be integrated into any network, as shown in Fig. 4. It can

adjust the weights of each channel and improves the attention to important channels, which is particularly beneficial in deep residual architectures. In this article, the SE block is directly integrated with the residual network to form the SE-ResNet architecture. This integration allows SE-ResNet to achieve improved performance in differential cryptanalysis, by making the network more sensitive to informative features and more robust to variations in input data.

# THE FRAMEWORK FOR DEVELOPING RELATED-KEY DIFFERENTIAL NEURAL DISTINGUISHERS TO SIMON AND SIMECK

The development of related-key differential neural distinguisher consists of four steps: differences selection, sample generation, network architecture design, distinguisher training and distinguisher evaluation, as shown in Fig. 5. In this section, we first introduce how to use a difference to construct the basic related-key differential neural distinguishers for SIMON and SIMECK from these steps. Subsequently, we introduce an advanced technique to construct the enhanced related-key differential neural distinguisher using a pair of distinct differences.

## Basic related-key differential neural distinguishers
### Differences selection

Selecting an appropriate plaintext difference $\Delta P$ and a master key difference $\Delta K$ for sample generation is a crucial step in the development of basic related-key differential neural distinguishers, since it significantly influences the features embodied within the samples. The study of *Gohr, Leander & Neumann (2022)*, *Bellini et al. (2023b)* indicates that the differences that can yield the ciphertext differences with high bias scores $b_r$ may be more suitable for constructing neural distinguishers. In the related-key scenario, the $r$-round exact bias score of ciphertext difference is defined as follows.

**Definition 5 (Exact bias score. *Gohr, Leander & Neumann (2022)*)** *For a cipher primitive* $E : \mathbb{F}_2^n \times \mathbb{F}_2^k \to \mathbb{F}_2^n$, *the* $r$-round bias score $b_r(\Delta P, \Delta K)$ *of the plaintext difference* $\Delta P \in \mathbb{F}_2^n$ *and master key difference* $\Delta K \in \mathbb{F}_2^k$ *is the sum of the biases of each bit position in the resulting ciphertext differences, i.e.,*

$$b_r(\Delta P, \Delta K) = \frac{1}{n} \sum_{j=0}^{n-1} \left| 0.5 - \frac{\sum_{X \in \mathbb{F}_2^n, K \in \mathbb{F}_2^k} \left( E_K(X) \oplus E_{K \oplus \Delta K}(X \oplus \Delta P) \right)_j}{2^{n+k}} \right|. \tag{7}$$

However, due to the immense computational demands posed by the exhaustive enumeration of all possible plaintexts and keys, computing the exact bias score is impractical. Therefore, we have to adopt more efficient methods to do this work. One promising approach is statistical sampling techniques, which is employed in *Gohr, Leander & Neumann (2022)*. By employing random sampling method, we could reduce the time and resources required for data collection and analysis while maintaining a high level of accuracy and reliability. By randomly selecting $t$
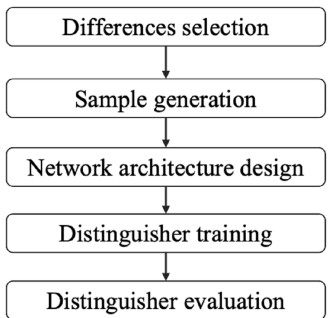

**Figure 5  The framework of basic and enhanced related-key differential neural distinguishers.**

samples from the plaintext and key space, we can obtain an approximate bias score $\widetilde{b}_r^t(\Delta P, \Delta K)$ as follow:

$$\widetilde{b}_r^t(\Delta P, \Delta K) = \frac{1}{n}\sum_{j=0}^{n-1}\left|0.5 - \frac{1}{t}\sum_{i=0}^{t-1}(E_{K_i}(X_i) \oplus E_{K_i \oplus \Delta K}(X_i \oplus \Delta P))_j\right|. \tag{8}$$

In addition, to mitigate the instance where certain differences have low bit bias in the initial few rounds but exhibit favorable bit bias in subsequent rounds, a practical strategy is to calculate the bias score from the initial round and adopt their weighted bias score as the final the final metric for evaluation. This approach can enhance the robustness of the differential evaluation. Specifically, the $R$-rounds weighted bias score $S_R(\Delta P, \Delta K)$ for a given plaintext difference $\Delta P$ and master key difference $\Delta K$ is the sum of the product of the number of rounds and their bias score. The mathematical expression is as follows:

$$S_R(\Delta P, \Delta K) = \sum_{r=1}^{R} r \times \widetilde{b}_r^t(\Delta P, \Delta K). \tag{9}$$

### Sample generation

The related-key differential neural distinguisher is a supervised binary classifier. Thus, its dataset consists of positive and negative samples, labeled as 1 and 0, respectively. The positive samples are obtained by encrypting the plaintext pairs using the key pairs that exhibit the plaintext difference $\Delta P$ and key difference $\Delta K$. In contrast, the negative samples are derived from encrypting the random plaintext pairs using the random key pairs.

Following the work of *Lu et al. (2024)*, we use eight ciphertext pairs with boosted data formats to train the related-key differential neural distinguishers for SIMON and SIMECK. Specifically, the $i$-th ($1 \leq i \leq 8$) $r$-round ciphertext pair $(C_l, C_r, C_l', C_r')_i$, derived from the $i$-th plaintext pair $(P, P')_i$ and key pair $(K, K')_i$, can be extended to $(\Delta_L^r, \Delta_R^r, C_l, C_r, C_l', C_r', \Delta_R^{r-1}, p\Delta_R^{r-2})_i$, denoted as $\Omega_i$, where

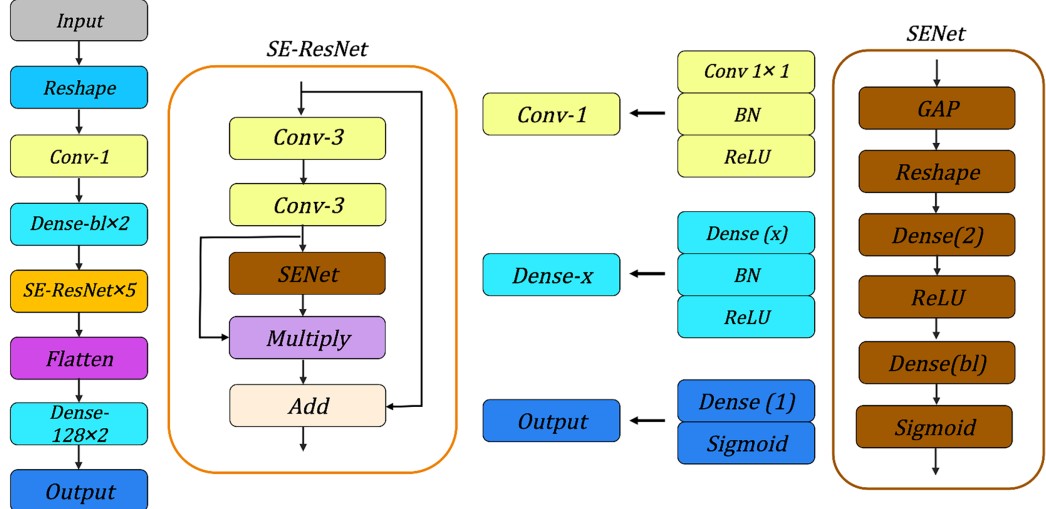

**Figure 6 Overview of neural network architectures.** BN, Batch Normalization; GAP, Global Average Pooling.

$$\begin{cases} \Delta_L^r = C_l \oplus C_l', \\ \Delta_R^r = C_r \oplus C_r', \\ f(x) = (x \lll \alpha) \odot (x \lll \beta) \oplus (x \lll \gamma), \\ \Delta_R^{r-1} = f(C_r) \oplus C_l \oplus f(C_r') \oplus C_l', \\ p\Delta_R^{r-2} = f(f(C_r) \oplus C_l) \oplus C_r \oplus f(f(C_r') \oplus C_l') \oplus C_r'. \end{cases} \tag{10}$$

The label $Y$ of the sample $(\Omega_1 \| \Omega_2 \| \ldots \| \Omega_s)$ can be expressed as

$$Y(\Omega_1 \| \Omega_2 \| \ldots \| \Omega_s) = \begin{cases} 1, & if \ P_i \oplus P_i' = \Delta P \ and \ K_i \oplus K_i' = \Delta K, \\ 0, & else. \end{cases} \tag{11}$$

### Network architecture

We evaluate the various neural network architectures for the `SIMON` and `SIMECK`, such as neural network architectures used in *Gohr (2019)*, *Bao et al. (2022)*, *Lu et al. (2024)* and *Zhang, Wang & Chen (2023)*, the architecture shown in Fig. 6 can achieve best accuracy under the same conditions. It consists of the following components:

- *Input Layer:* For the `SIMON` and `SIMECK` with a block length of *bl*, the input of neural network is a tensor with a shape of $(8 \times bl \times 4, 1)$.
- *Reshape Layer:* This layer transforms the input tensor into a new shape of $(8, bl \times 8)$ to enhance the feature extraction for subsequent convolutional layers.
- *Conv-1:* A convolutional layer with *bl* convolutional kernels of size 1, followed by a batch normalization layer and a ReLU activation function.
- *Dense bl × 2:* Two dense layers implemented sequentially to process the features extracted from the *Conv-1*. Each dense layer consists of *bl* neurons followed by a batch normalization layer and a ReLU activation function.

- *SE-ResNet × 5:* A sequence of five SE-ResNet layers. Each SE-ResNet integrates the ResNet and SENet architectures and contains two convolutional layers with $3 \times 3$ kernels for feature extraction, followed by a batch normalization layer, a ReLU activation function, and a Squeeze-and-Excitation module. The features from different layers are merged by *Multiply* and *Add* operations.
- *Flatten:* This layer flattens the multi-dimensional output from the SE-ResNet layer into a one-dimensional tensor.
- *Dense-128 × 2:* Two fully connected layers with 128 neurons are used to connect all the features and send the output to the Sigmoid classifier in the subsequent layer.
- *Output:* The final layer of the neural network is responsible for generating the final prediction result.

### Training and evaluation

The training process of a related-key differential neural distinguisher can be divided into two phases: the offline phase and the online phase. During the offline phase, the attacker aims to train a neural network that can effectively distinguish between positive and negative samples. To achieve this, the attacker first generates training samples and validation samples using selected plaintext difference $\Delta P$ and master key difference $\Delta K$. The training samples are used to train the neural network, while the validation samples are used to evaluate the recognition ability of the neural network. Ultimately, we can determine whether we have successfully constructed an effective neural distinguisher based on whether its accuracy surpasses the threshold of 0.5.

In the online phase, the neural distinguisher trained in the offline phase is employed to distinguish the ciphertext data generated by a block cipher or a random function. If the score of more than half of the samples exceeds 0.5, we consider the ciphertext data comes from the block cipher. Otherwise, these data are considered to originate from the random function.

### Parameter setting

The number of training samples and validation samples used in this article is $2 \times 10^7$ and $2 \times 10^6$. In addition, we set the number of epochs to 120, and each epoch contains multiple batches, each containing 30,000 samples. In order to adjust the learning rate more efficiently, we adopt the cyclic learning rate. Specifically, for the $i$-th epoch, its learning rate $l_i$ is dynamically calculated by $l_i = a + \frac{(n-i) \bmod (n+1)}{n} \times (b - a)$, where $a = 0.0001$, $b = 0.003$, and $n = 29$. Moreover, we choose Adam (*Kingma & Ba, 2014*) as the optimizer and Mean Squared Error (MSE) as the loss function. To prevent the model from overfitting, we use L2 regularization with the parameter $c$ of 0.00001.

## Enhanced related-key differential neural distinguishers
### Motivation
*Benamira et al. (2021)* found that Gohr's neural distinguisher showed a superior recognition ability for the ciphertext pairs exhibiting truncated differences with high

probability in the last two rounds, suggesting a potential understanding and learning of differential-linear characteristics in the ciphertext pairs. Subsequently, *Gohr, Leander & Neumann (2022)* expanded their study to five different block, including SIMON, Speck (*Beaulieu et al., 2015*), Skinny (*Beierle et al., 2016*), PRESENT *Bogdanov et al. (2007)*, Katan (*De Canniere, Dunkelman & Knežević, 2009*), and ChaCha (*Bernstein, 2008*). Notably, their research highlights the close connection between the accuracy of the neural distinguisher and the mean absolute distance of the ciphertext differential distribution and the uniform distribution. In light of these investigations, we enhance the basic differential neural distinguisher by using two distinct non-zero plaintext differences and master key differences, symbolically represented as $(\Delta P, \Delta P', \Delta K, \Delta K')$.

The primary rationale behind selecting two input differences instead of one or more stems from the objective of minimizing conflicts among the output differences arising from positive and negative samples. When an input difference is chosen, as the number of rounds increases, some output differences will tend to be uniformly distributed due to the inherent confusion and diffusion properties of the block cipher. This poses a great challenge for the neural network to distinguish them from the uniformly distributed negative samples. However, if the negative samples are generated from another good difference, the mean absolute distance between the positive and negative samples may become more significant, which can allow the neural network to distinguish them more effectively. There are two reasons for limiting the number of input differences to two rather than more: firstly, the input differences that can maintain their unique distribution across several rounds are rare; secondly, an increase in the variety of ciphertext data may heighten the likelihood of collisions.

### Differences selection

To develop an efficient and enhanced neural distinguisher, $(\Delta P, \Delta P', \Delta K, \Delta K')$ needs to satisfy two pivotal requirements. Firstly, they must exhibit a favorable weighted bias score after several rounds, ensuring that the resulting ciphertext data possess distinct and discernible features. This can be straightforwardly accomplished by adopting the differential evaluation scheme detailed in "Basic Related-Key Differential Neural Distinguishers". Second, the disparity between the ciphertext data derived from the input differences $(\Delta P, \Delta K)$ and $(\Delta P', \Delta K')$ should be maximized, thereby ensuring that there are sufficient features for the neural network to leverage during the learning process.

Inspired by the role of weighted bias scores, we try to directly utilize their relative weighted bias scores, denoted as $S_R(\Delta P, \Delta P', \Delta K, \Delta K')$, as a rough metric to evaluate the suitability of $(\Delta P, \Delta P', \Delta K, \Delta K')$ for building the enhanced neural distinguishers, where

$$\widetilde{b}_r^t(\Delta P, \Delta P', \Delta K, \Delta K') = \frac{1}{n}\sum_{j=0}^{n-1}\left|\frac{1}{t}\sum_{i=0}^{t-1}\left(E_{K_i \oplus \Delta K}(X_i \oplus \Delta P) - E_{K_i \oplus \Delta K'}(X_i \oplus \Delta P')\right)_j\right|. \quad (12)$$

$$S_R(\Delta P, \Delta P', \Delta K, \Delta K') = \sum_{r=1}^{R} r \times \widetilde{b}_r^t(\Delta P, \Delta P', \Delta K, \Delta K'). \quad (13)$$

However, the outcomes are disappointing, primarily due to the fact that the relative weighted bias scores among all combinations derived from two input differences with weighted high bias scores have a high degree of similarity.

Fortunately, the differences that have high weighted bias scores are generally scarce. For a set of $m$ input differences, the total number of potential combinations is $\frac{m \times (m-1)}{2}$. Consequently, when $m$ is small, the exhaustive approach that compares all potential combinations to identify the optimal one is feasible. Nonetheless, as the value of $m$ increases, the number of combinations grows rapidly. Specifically, when $m$ is 32, it is a daunting task to train 496 neural distinguishers. Given that the training of a single neural distinguisher takes about an hour and a half, the aggregate time required for this task approximating 31 days, which is impractical and and unacceptable for most researchers. Therefore, the adoption of a more efficient and targeted strategy for selecting promising combinations becomes imperative.

An available greedy strategy is to fix $(\Delta P, \Delta K)$ as the optimal or top-ranked input difference that can be used to construct the most effective basic neural distinguisher. Subsequently, $(\Delta P', \Delta K')$ is chosen from the remaining differences with good weighted bias score. This strategy can ensure that the ciphertext data generated with $(\Delta P, \Delta K)$ have discernible and distinctive features. In this article, we adopt the exhaustive approach for `SIMON32/64` and `SIMON32/64`. For the remaining variants, we adopt this greedy strategy to speed up the process of differences selection.

### Sample generation

The sample generation for enhanced neural distinguisher is different from method outlined for the basic neural distinguisher in "Basic Related-Key Differential Neural Distinguishers". For the enhanced neural distinguisher, the positive and negative samples are ciphertext data generated from the plaintext pairs and key pairs with the differences $(\Delta P, , \Delta K)$ and $(\Delta P', \Delta K')$. The label of a sample $(\Omega_1 \| \Omega_2 \| \dots \| \Omega_s)$ is represented as

$$Y(\Omega_1\|\Omega_2\|\dots\|\Omega_s) = \begin{cases} 1, & if \ P_i \oplus P'_i = \Delta P \ and \ K_i \oplus K'_i = \Delta K, \\ 0, & if \ P_i \oplus P'_i = \Delta P' \ and \ K_i \oplus K'_i = \Delta K'. \end{cases} \tag{14}$$

The neural network architecture and the process of training and evaluation remain consistent with that in "Basic Related-Key Differential Neural Distinguishers".

## RELATED-KEY DIFFERENTIAL NEURAL DISTINGUISHERS FOR ROUND-REDUCED SIMON AND SIMECK

In this section, we adopt the framework and strategies in "The Framework for Developing Related-Key Differential Neural Distinguishers to SIMON and SIMECK" to develop the basic and enhanced related-key differential neural distinguishers for SIMON and SIMECK.

### Differences selection for SIMON

#### The differences with Hamming weights of 1 and 2

For a block cipher with block length $bl$ and key length $kl$, the number of input differences we need to evaluate is $2^{bl+kl}$. Even for the smallest variants, *i.e.*, SIMON32/64 and

SIMECK32/64, the number of differences that need to be evaluated reaches $2^{96}$, which would take a lot of time. Therefore, we first evaluate the weighted bias scores for all the differences with Hamming weights of 1 and 2.

For the 8-round SIMON32/64, there are 16 input differences with weighted bias scores around 11.0, which are $\Delta P = (0\times0, 0\times1 \lll i), \Delta K = (0\times0, 0\times0, 0\times0, 0\times1 \lll i)$, $i \in [0, 15]$. This is followed by another 16 input differences with a weighted bias score of about 10.8, specified as $\Delta P = (0\times0, 0\times21 \lll i), \Delta K = (0\times0, 0\times0, 0\times0, 0\times21 \lll i)$, $i \in [0, 15]$. The score for all remaining input differences with Hamming weights of 1 and 2 is less than 10.00.

For the 8-round SIMON48/96, there are 24 input differences with a Hamming weight of 1 that have a weighted bias score between 15.3 and 14.4: $\Delta P = (0\times0, 0\times1 \lll i)$, $\Delta K = (0\times0, 0\times0, 0\times0, 0\times1 \lll i)$, $i \in [0, 23]$. For differences with a Hamming weight of 2, only 11 input differences yield weighted bias scores greater than 14.4. They are $\Delta P = (0\times0, 0\times41000 \lll i)$, $\Delta K = (0\times0, 0\times0, 0\times0, 0\times41000 \lll i)$, $i \in [0, 6]$, $\Delta P = (0\times0, 0\times21000 \lll i)$, $\Delta K = (0\times0, 0\times0, 0\times0, 0\times21000 \lll i)$, $i \in [0, 2]$, and $\Delta P = (0\times0, 0\times30000)$, $\Delta K = (0\times0, 0\times0, 0\times0, 0\times30000)$.

For the 8-round SIMON64/128, there are 32 differences with a Hamming weight of 1 that exhibit scores around 13.4. These differences are denoted as $\Delta P = (0\times0, 0\times1 \lll i)$, $\Delta K = (0\times0, 0\times0, 0\times0, 0\times1 \lll i)$, $i \in [0, 31]$. After that, there are 32 differences with Hamming weight 2 that have scores close to 12.6 or 12.5, which are $\Delta P = (0\times0, 0\times21 \lll i)$, $\Delta K = (0\times0, 0\times0, 0\times0, 0\times21 \lll i)$, $i \in [0, 31]$, and $\Delta P = (0\times0, 0\times41 \lll i)$, $\Delta K = (0\times0, 0\times0, 0\times0, 0\times41 \lll i)$, $i \in [0, 31]$, respectively. The scores for all remaining differences are below 12.2.

### Structural features of SIMON

For SIMON32/64, SIMON48/96, and SIMON64/128, the input differences with high weighted bias scores are those with the structure $\Delta P = (0\times0, \Delta X)$ and $\Delta K = (0\times0, 0\times0, 0\times0, \Delta X)$. By analyzing the propagation process of the difference in SIMON, we can find that the plaintext differences and key differences cancel each other out in the first round. In the next three rounds, both plaintext difference and key difference are zero. Only in the fifth round, the key difference $\Delta X$ is re-injected, and the plaintext difference is still zero. The detailed differential propagation process is given in Table 3. This is easily verified by analyzing the transformations of the plaintext difference and the key difference in the round function and the round key.

### The differences with a Hamming weight greater than 2

Based on the structural feature of SIMON, for differences with a weight greater than 2, we only consider the differences with a structure of $\Delta P = (0\times0, \Delta X)$ and $\Delta K = (0\times0, 0\times0, 0\times0, \Delta X)$. For 8-round SIMON32/64, there are only 32 differences with Hamming weights of 3 that have weighted bias scores greater than 10.0. Specifically, they are $\Delta P = (0\times0, 0\times43/0\times421 \lll i)$, $\Delta K = (0\times0, 0\times0, 0\times0, 0\times43/0\times421 \lll i)$, $i \in [0, 15]$, with scores between 10.7 and 10.3. For the 8-round SIMON48/96 and

**Table 3 The related-key differential characteristic of `SIMON` with four key words.**

| Round | $\Delta P_r$ | $\Delta K_r$ |
|---|---|---|
| 1 | $(0\times 0, \Delta X)$ | $\Delta X$ |
| 2 | $(0\times 0, 0\times 0)$ | $0\times 0$ |
| 3 | $(0\times 0, 0\times 0)$ | $0\times 0$ |
| 4 | $(0\times 0, 0\times 0)$ | $0\times 0$ |
| 5 | $(0\times 0, 0\times 0)$ | $\Delta X$ |

`SIMON64/128`, the weighted bias scores for all differences with a Hamming weight greater than two are less than 14.4 and 12.2, respectively.

### Differences selection for `SIMECK`

#### *The differences with Hamming weights of 1 and 2*

Following the experiments on `SIMON`, we first explore the applicability of the input differences with Hamming weights of 1 and 2 in constructing neural distinguishers for `SIMECK`. For 10-round `SIMECK32/64`, 16 differences with a Hamming weight of 1, denoted as $\Delta P = (0\times 0, 0\times 1 \lll i)$, $\Delta K = (0\times 0, 0\times 0, 0\times 0, 0\times 1 \lll i)$, $i \in [0, 15]$, achieve the optimal weighted bias score around 16.3. Then there are 32 differences with Hamming weight of 2, $\Delta P = (0\times 0, 0\times 3/0\times 11 \lll i)$, $\Delta K = (0\times 0, 0\times 0, 0\times 0, 0\times 3/0\times 11 \lll i)$, $i \in [0, 15]$, with scores greater than 13.0. The rest of the differences are scored below 13.0.

For the 12-round `SIMECK48/96`, there are 24 differences with a Hamming weight of 1, $\Delta P = (0\times 0, 0\times 1 \ll i)$, $\Delta K = (0\times 0, 0\times 0, 0\times 0, 0\times 1 \ll i)$, $i \in [0, 23]$, that have a weighted bias score between 30.4 and 26.6. For differences with a Hamming weight of 2, there are 33 differences with scores greater than or equal to 26.6. They are $\Delta P = (0\times 0, 0\times 30 \ll i)$, $\Delta K = (0\times 0, 0\times 0, 0\times 0, 0\times 30 \ll i)$, $i \in [0, 12]$, $\Delta P = (0\times 0, 0\times 220 \ll i)$, $\Delta K = (0\times 0, 0\times 0, 0\times 0, 0\times 220 \ll i)$, $i \in [0, 8]$, $\Delta P = (0\times 0, 0\times 140 \ll i)$, $\Delta K = (0\times 0, 0\times 0, 0\times 0, 0\times 140 \ll i)$, $i \in [0, 6]$, and $\Delta P = (0\times 0, 0\times 480 \ll i)$, $\Delta K = (0\times 0, 0\times 0, 0\times 0, 0\times 480 \ll i)$, $i \in [0, 3]$. The scores of all remaining differences are all less than 26.5.

For the 15-round `SIMECK64/128`, the best weighted bias score around 30.1 is achieved by 32 differences with a Hamming weight of 1, which are $\Delta P = (0\times 0, 0\times 1 \lll i)$, $\Delta K = (0\times 0, 0\times 0, 0\times 0, 0\times 1 \lll i)$, $i \in [0, 31]$. Then there are 32 differences, $\Delta P = (0\times 0, 0\times 3 \lll i)$, $\Delta K = (0\times 0, 0\times 0, 0\times 0, 0\times 3 \lll i)$, $i\ in[0, 31]$, with scores close to 26.7. All the other differences have scores below 26.0.

#### *Structural features of `SIMECK`*

Similar to `SIMON`, for all variants of `SIMECK`, the input differences that exhibit good weighted bias scores adhere to the format: $\Delta P = (0\times 0, \Delta X)$ and $\Delta K = (0\times 0, 0\times 0, 0\times 0, \Delta X)$. This is also due to the fact, as shown in Table 4, that the plaintext difference and key difference cancel each other out in the first round, and in the subsequent three rounds, both the plaintext difference and key difference are zero. It is not

**Table 4 The related-key differential characteristic of SIMECK.**

| Round | $\Delta P_r$ | $\Delta K_r$ |
|---|---|---|
| 1 | $(0 \times 0, \Delta X)$ | $\Delta X$ |
| 2 | $(0 \times 0, 0 \times 0)$ | $0 \times 0$ |
| 3 | $(0 \times 0, 0 \times 0)$ | $0 \times 0$ |
| 4 | $(0 \times 0, 0 \times 0)$ | $0 \times 0$ |
| 5 | $(0 \times 0, 0 \times 0)$ | $\Delta X'$ |

until the fifth round that the key difference $\Delta X'$, resulting from the $\odot$ operation of $\Delta K_r \lll \alpha$ and $\Delta K_r \lll \beta$, is reintroduced.

### *The differences with a Hamming weight greater than 2*

For the 10-round SIMECK32/64 and 15-round SIMECK64/128, none of the differences with a Hamming weight of more than two yields a weighted bias score above 12.5 and 24.5, respectively. For 12-round SIMECK48/96, there are only three differences with a Hamming weight of three that have a score of 26.8, which are $\Delta P = (0 \times 0, 0 \times 700 / 0 x e 00 / 0 \times 2300)$, $\Delta K = (0 \times 0, 0 \times 0, 0 \times 0, 0 \times 700 / 0 x e 00 / 0 \times 2300)$. The scores for all remaining differences with a Hamming weight of three or higher are all below 26.6.

### Basic related-key differential neural distinguishers

For the SIMON32/64, the 16 most effective 13-round related-key differential neural distinguishers are trained using the candidate differences $\Delta P = (0 \times 0, 0 \times 21 \lll i)$, $\Delta K = (0 \times 0, 0 \times 0, 0 \times 0, 0 \times 21 \lll i)$ where i ranges from 0 to 15. Their accuracy is $0.543 \pm 0.002$, while it is $0.525 \pm 0.005$ for the distinguishers built from the candidate differences $\Delta P = (0 \times 0, 0 \times 1 \lll i)$, $\Delta K = (0 \times 0, 0 \times 0, 0 \times 0, 0 \times 1 \lll i)$, $i \in [0, 15]$. The best 13-round neural distinguisher is constructed by $\Delta P = (0 \times 0, 0 \times 2004)$, $\Delta K = (0 \times 0, 0 \times 0, 0 \times 0, 0 \times 2004)$ with an accuracy of 0.545. Its 12-round neural distinguisher achieves an accuracy of 0.678. Compared with the related-key differential neural distinguisher in *Lu et al. (2024)*, our differential selection strategy enables us to yield the superior distinguisher, as shown in Table 1.

For SIMON48/96, the best 13-round related-key differential neural distinguisher with an accuracy of 0.650 is constructed with $\Delta P = (0 \times 0, 0 \times 200000)$ and $\Delta K = (0 \times 0, 0 \times 0, 0 \times 0, 0 \times 200000)$. Its 12-round neural distinguisher can achieve an accuracy of 0.993. For the remaining 23 candidate differences with a Hamming weight of 1, the accuracy of their 13-round neural distinguishers is between 0.640 to 0.650. In contrast, when the candidate differences with Hamming weight 2 in "Differences Selection for SIMON" is adopted, the highest accuracy is only 0.593, which is lower than that of 24 candidate differences with a Hamming weight of 1. Moreover, the three candidate differences with a Hamming weight of three could not construct an effective neural distinguisher for 13 rounds.

For SIMON64/128, the optimal 14-round related-key differential neural distinguisher is constructed using $\Delta P = (0 \times 0, 0 \times 100000)$ and $\Delta K = (0 \times 0, 0 \times 0, 0 \times 0, 0 \times 100000)$ with an accuracy of 0.580. The accuracy of its 13-round neural distinguisher is 0.840. In

addition, the neural distinguishers built from the other 31 candidate differences with a Hamming weight of one exhibit accuracy between 0.577 and 0.580. There are no valid 14-round neural distinguishers achieved when using the candidate differences with a Hamming weight of two in "Differences Selection for SIMON".

For SIMECK, the maximum number of rounds that can be constructed for related-key differential neural distinguishers is 15 for SIMECK32/64, 19 for SIMECK48/96, and 22 for SIMECK64/128. Their optimal neural distinguishers are constructed using $\Delta P = (0\times0, 0\times10/0\times2/0\times200000)$ and $\Delta K = (0\times0, 0\times0, 0\times0, 0\times10/0\times2/0\times200000)$ with an accuracy of 0.547, 0.516, and 0.519, respectively. The accuracies of these neural distinguishers from the previous round are 0.668, 0.551, and 0.552, respectively. The neural distinguishers constructed from other candidate differences with a Hamming weight of one have an accuracy very close to the best neural distinguisher above, with a maximum deviation of only 0.002. The candidate differences with Hamming weights greater than two fail to construct effective neural distinguishers with the maximum number of rounds.

### Enhanced related-key differential neural distinguishers

For the SIMON32/64 and SIMECK32/64, we use all possible combinations of the superior candidate differences $\Delta P = (0\times0, 0\times21/0\times1 \lll i)$ and $\Delta K = (0\times0, 0\times0, 0\times0, 0\times21/0\times1 \lll i)$, $i \in [0, 15]$, to construct the related-key differential neural distinguisher. For SIMON32/64, there are five different $(\Delta P, \Delta P', \Delta K, \Delta K')$ that can yield the 13-round related-key differential neural distinguisher with an accuracy of 0.567. They are

$$\begin{cases} \Delta P = (0\times0, 0\times801/0\times42/0\times2100/2004/2100), \Delta K = (0\times0, 0\times0, 0\times0, 0\times100000/0\times2/0\times200000), \\ \Delta P' = (0\times0, 0\times1002/0\times84/0\times1080/1002/4200), \Delta k' = (0\times0, 0\times0, 0\times0, 0\times400000/0\times80000/0\times200). \end{cases}$$

For the first two instances, the accuracy of their 12-round neural distinguisher is 0.740, while it is 0.738 for the remaining three instances.

For SIMON48/96, SIMON64/128, SIMECK48/96, and SIMECK64/128, we consider combinations of the best differences in Table 5 and the remaining candidate differences of $\Delta P = (0\times0, 0\times1 \lll i)$ and $\Delta K = (0\times0, 0\times0, 0\times0, 0\times1 \lll i)$, $i \in [0, 15]$ to accelerate the construction of our enhanced neural distinguishers. Specifically, for SIMON48/96, there are three pairs of differences that can yield 12-round and 13-round related-key differential neural distinguishers with accuracies of 0.997 and 0.696, respectively. These pairs are $\Delta P = (0\times0, 0\times200000)$ and $\Delta K = (0\times0, 0\times0, 0\times0, 0\times2000)$ together with $\Delta P' = (0\times0, \Delta)$ and $\Delta K' = (0\times0, 0\times0, 0\times0, \Delta)$, where $\Delta \in [0\times400000, 0\times100000, 0\times40]$. For SIMON64/128, SIMECK48/96, and SIMECK64/128, only one pair of differences can construct 14-round, 19-round, and 22-round related-key neural distinguishers with accuracies of 0.618, 0.523, and 0.526, respectively. They are $\Delta P = (0\times0, 0\times100000/0\times2/0\times200000)$, $\Delta K = (0\times0, 0\times0, 0\times0, 0\times100000/0\times2/0\times200000)$, $\Delta P' = (0\times0, 0\times400000/0\times80000/0\times200)$, and $\Delta K' = (0\times0, 0\times0, 0\times0, 0\times400000/0\times80000/0\times200)$. The accuracies of 13-round, 18-round, and 21-round neural distinguishers for these pairs are 0.916, 0.572, and 0.572, respectively, as shown in Table 6.

**Table 5  The basic related-key differential neural distinguishers for SIMON and SIMECK.**

| Cipher | Round | ΔP | ΔK | Acc | TPR | TNR |
|---|---|---|---|---|---|---|
| SIMON32/64 | 12 | (0×0, 0×2004) | (0×0, 0×0, 0×0, 0×2004) | 0.678 | 0.685 | 0.671 |
| | 13 | (0×0, 0×2004) | (0×0, 0×0, 0×0, 0×2004) | 0.545 | 0.537 | 0.552 |
| SIMON48/96 | 12 | (0×0, 0×200000) | (0×0, 0×0, 0×0, 0×200000) | 0.993 | 0.999 | 0.986 |
| | 13 | (0×0, 0×200000) | (0×0, 0×0, 0×0, 0×200000) | 0.650 | 0.660 | 0.640 |
| SIMON64/128 | 13 | (0×0, 0×100000) | (0×0, 0×0, 0×0, 0×100000) | 0.840 | 0.834 | 0.845 |
| | 14 | (0×0, 0×100000) | (0×0, 0×0, 0×0, 0×100000) | 0.580 | 0.575 | 0.585 |
| SIMECK32/64 | 14 | (0×0, 0×10) | (0×0, 0×0, 0×0, 0×10) | 0.668 | 0.640 | 0.695 |
| | 15 | (0×0, 0×10) | (0×0, 0×0, 0×0, 0×10) | 0.547 | 0.524 | 0.570 |
| SIMECK48/96 | 18 | (0×0, 0×2) | (0×0, 0×0, 0×0, 0×2) | 0.551 | 0.456 | 0.646 |
| | 19 | (0×0, 0×2) | (0×0, 0×0, 0×0, 0×2) | 0.516 | 0.411 | 0.611 |
| SIMECK64/128 | 21 | (0×0, 0×200000) | (0×0, 0×0, 0×0, 0×200000) | 0.552 | 0.413 | 0.691 |
| | 22 | (0×0, 0×200000) | (0×0, 0×0, 0×0, 0×200000) | 0.519 | 0.374 | 0.663 |

**Table 6  The enhanced related-key differential neural distinguishers for SIMON and SIMECK.**

| Cipher | ΔP/ΔP′ | ΔK/ΔK′ | Round | Acc | TPR | TNR |
|---|---|---|---|---|---|---|
| SIMON32/64 | (0×0, 0×801) | (0×0, 0×0, 0×0, 0×801) | 12 | 0.740 | 0.729 | 0.750 |
| | (0×0, 0×1002) | (0×0, 0×0, 0×0, 0×1002) | 13 | 0.567 | 0.564 | 0.570 |
| SIMON48/96 | (0×0, 0×200000) | (0×0, 0×0, 0×0, 0×200000) | 12 | 0.997 | 0.998 | 0.996 |
| | (0×0, 0×400000) | (0×0, 0×0, 0×0, 0×400000) | 13 | 0.696 | 0.698 | 0.695 |
| SIMON64/128 | (0×0, 0×100000) | (0×0, 0×0, 0×0, 0×100000) | 13 | 0.916 | 0.910 | 0.922 |
| | (0×0, 0×400000) | (0×0, 0×0, 0×0, 0×400000) | 14 | 0.618 | 0.596 | 0.639 |
| SIMECK32/64 | (0×0, 0×80) | (0×0, 0×0, 0×0, 0×80) | 14 | 0.730 | 0.722 | 0.738 |
| | (0×0, 0×2000) | (0×0, 0×0, 0×0, 0×2000) | 15 | 0.568 | 0.553 | 0.582 |
| SIMECK48/96 | (0×0, 0×2) | (0×0, 0×0, 0×0, 0×2) | 18 | 0.572 | 0.572 | 0.572 |
| | (0×0, 0×80000) | (0×0, 0×0, 0×0, 0×80000) | 19 | 0.523 | 0.527 | 0.518 |
| SIMECK64/128 | (0×0, 0×200000) | (0×0, 0×0, 0×0, 0×200000) | 21 | 0.572 | 0.580 | 0.563 |
| | (0×0, 0×200) | (0×0, 0×0, 0×0, 0×200) | 22 | 0.526 | 0.523 | 0.529 |

## Comparison and discussion

In this section, we first evaluate the differences with Hamming weights of 1 and 2 for SIMON and SIMECK, using weight bias scores. Then, we further evaluate the differences with Hamming weights greater than two based on the structural features of SIMON and SIMECK. Compared with the exhaustive approach of training a neural distinguisher for each difference in *Lu et al. (2024)*, our scheme is more efficient.

Using these differences, we can obtain 13-round basic related-key differential neural distinguishers, exhibiting superior accuracy than that in *Lu et al. (2024)*, for SIMON32/64. The accuracy of this basic neural distinguisher can be improved from 0.526 to 0.545 due to the effectiveness of our difference selection strategy. For the remaining variants, we can also obtain the basic related-key differential neural distinguishers with the same accuracy

as that in *Lu et al. (2024)*, as shown in Table 1. In addition, we obtain multiple basic related-key differential neural distinguishers that have the same or similar accuracy as the best distinguisher. All these results illustrate the effectiveness and usability of our proposed strategy for difference selection.

When constructing the enhanced related-key differential neural distinguishers using our method, all the enhanced differential neural distinguishers achieve higher accuracy than the basic related-key differential neural distinguishers for both `SIMON` and `SIMECK`. Compared with the results in *Lu et al. (2024)*, our neural distinguishers all achieve different degrees of improvement in accuracy, as shown in Table 1. Specifically, the accuracy of the 13-round `SIMON32/64`, 13-round `SIMON48/96`, and 14-round `SIMON64/128` is increased from 0.545, 0.650, and 0.580 to 0.567, 0.696, and 0.618, respectively. Similarly, the neural distinguishers for the 15-round `SIMECK32/64`, 19-round `SIMECK48/96`, and 22-round `SIMECK64/128` is also demonstrated notable improvements in accuracy, with increases from 0.547, 0.516, and 0.519 to 0.568, 0.523, and 0.526, respectively. These results collectively underscore the effectiveness and robustness of our proposed scheme for constructing the enhanced related-key differential neural distinguishers.

## CONCLUSIONS AND FUTURE WORK

In this article, we first establish a comprehensive framework to construct basic related-key differential neural distinguishers for the `SIMON` and `SIMECK`. To choose an appropriate difference to construct this distinguisher, we utilize weighted bias scores to assess the applicability of various differences, which speeds up the process of difference selection and evaluation.

Moreover, we introduce an innovative method that incorporates two distinct differences into the neural distinguisher instead of a differences, which can result in the more robust and effective neural distinguishers. Compared with the distinguishers in *Lu et al. (2024)*, we successfully improve the accuracy of the related-key differential neural distinguisher for both `SIMON` and `SIMECK` as shown in Table 1.

Furthermore, we envision several promising directions for future research. Firstly, our framework can be easily extended to other block ciphers, which assists in evaluating the security of other block ciphers. Secondly, the integration of advanced neural network architectures and training techniques could yield even more powerful neural distinguishers.

### Funding

This work is supported by the National Key Research and Development Program of China (2022YFB2701900), the National Natural Science Foundation of China (Nos. 62472172, 62072181), the Open Fund of Advanced Cryptography and System Security Key Laboratory of Sichuan Province (Grant No. SKLACSS-202301), and the Shanghai Trusted Industry Internet Software Collaborative Innovation Center. The funders had no role in

study design, data collection and analysis, decision to publish, or preparation of the manuscript.

## Grant Disclosures

The following grant information was disclosed by the authors:
National Key Research and Development Program of China: 2022YFB2701900.
National Natural Science Foundation of China: 62472172, 62072181.
Open Fund of Advanced Cryptography and System Security Key Laboratory of Sichuan Province: SKLACSS-202301.

## Competing Interests

The authors declare that they have no competing interests.

## Author Contributions

- Gao Wang conceived and designed the experiments, performed the experiments, analyzed the data, performed the computation work, prepared figures and/or tables, authored or reviewed drafts of the article, and approved the final draft.
- Gaoli Wang conceived and designed the experiments, prepared figures and/or tables, authored or reviewed drafts of the article, and approved the final draft.

## Data Availability

The raw data and code are available at Zenodo: differentialdistinguisher. (2024). differentialdistinguisher/Enhanced-related-key-differential-neural-distinguishers-for-simon-and-simeck-block-ciphers: First release (v1.0.0). Zenodo. https://doi.org/10.5281/zenodo.11178441.

## Supplemental Information

Supplemental information for this article can be found online at http://dx.doi.org/10.7717/peerj-cs.2566#supplemental-information.

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
