# Peer review of "Enhanced related-key differential neural distinguishers for SIMON and SIMECK block ciphers"

_PeerJ Computer Science, doi:10.7717/peerj-cs.2566_

## Round 0.1 · original submission · Major Revisions

Please highlight your main concern with this manuscript. How do you want to contribute to the literature with your results? In a theoretical or methodological way, how does your distinguisher differ from currently proposed ideas? Check the current literature and provide a comprehensive comparison with them. Language needs proofreading. A comparison and discussion section is required. Check the current literature and provide information and differentiations of your approach from the current literature. A detailed network architecture is proposed as a distinguisher by providing a working flow of a scheme.

**Language Note:** The review process has identified that the English language must be improved. PeerJ can provide language editing services - please contact us at [email protected] for pricing (be sure to provide your manuscript number and title). Alternatively, you should make your own arrangements to improve the language quality and provide details in your response letter. – PeerJ Staff

·

Basic reporting

This paper primarily investigates how to enhance related-key differential neural distinguishers for SIMON and SIMECK ciphers. The structure of the paper is well-organized, and the presentation is clear and concise. The references are comprehensive and well-cited. Additionally, the authors have provided the source code, which significantly enhances the transparency, authenticity, and reliability of the data and results presented.

Experimental design

This paper investigates the resistance of the SIMON and SIMECK ciphers against intelligent related-key differential cryptanalysis by integrating neural networks with related-key differential cryptanalysis. By varying the input differences, the authors achieve a higher accuracy related-key neural distinguisher for the SIMON32/64. Furthermore, by modifying the distinguishing tasks, they present related-key neural distinguishers with improved accuracy for both the SIMON and SIMECK algorithms. This paper reflects a substantial amount of work and is presented with clarity.

Validity of the findings

The authors achieve improvements in accuracy for related-key neural distinguishers by varying input differences and modifying distinguishing tasks for both ciphers. The methodology is detailed, enabling readers to follow the authors' reasoning. Overall, the paper reflects substantial effort, with a clear presentation.

Additional comments

1. The citation format throughout the paper is incorrect, which disrupts the flow of reading. Specifically, the references are written in text rather than using the customary numerical citations.

2. Some formulas in the paper lack proper ending punctuation, and certain formulas span an entire line without being centered. For example, formula (4) on page 6.

3. In Table 1, aside from SIMON32/64, the RKND results do not show any improvement, making their inclusion in the table seem redundant.

4. Although the RKND' results show improvement, have the authors considered the differences between this distinguishing task and traditional distinguishing tasks, and whether key recovery attacks are feasible?

Reviewer 2 ·

Basic reporting

The paper employs weighted bias scores to select input differences and trains a related-key differential neural distinguisher using two input differences. This approach enhances the accuracy of the differential neural distinguisher compared to the method proposed by Lu et al.

Experimental design

no comments

Validity of the findings

The use of two different input differences provides more information in the ciphertext, which naturally results in an improved distinguisher.

Additional comments

- In Formula 10, the design of positive and negative samples differs from previous differential neural distinguishers, with the negative samples possibly being non-random. Therefore, it is inappropriate to compare the results of this paper with those of Lu et al., as they do not address the same task.

- The authors mention that \((\Delta P', \Delta K')\) is chosen from the remaining differences in line 244, but no specific values are provided in the subsequent text.

- While the authors use two input differences, the trained distinguisher may be challenging to apply in key recovery attacks.

Minor comments:
- A blank line should be added before certain non-indented paragraphs to improve the layout, such as those on lines 150, 153, 254, and 276.

- There is a issue with the reference on line 224 that need to be addressed.

- The term should be written as Simon32/64, not Simon3264.

Reviewer 3 ·

Basic reporting

This paper provides a framework to construct related-key differential neural distinguishers for SIMON and SIMECK. To efficiently choose the input differences for the related-key
differential neural distinguisher, they introduce a method to approximate the validity of various input
differences. The work is interesting. This paper is also well organized, well written, clear and unambiguous.

some questions:

1. Line 146, such as image recognitionChauhan et al.?? the citations in this paper should be given in a better manner.
2. Please give a CNN net structure is Section 2.4, the readers can understand it better.
3. the equations should be better aligned.
4. Could you please give a figure for your framework in Section 3?
5. Figure 2, please explain why you choose such a network?
6. information of some references is lost, e.g., line 399.

Experimental design

The experiments are well given and robust.

Validity of the findings

The results well illustrate the main idea and the contribution of this paper.

---

## Round 0.2 · Major Revisions

In the response letter, please answer the reviewers' requests individually and include any additions based on the reviewers' comments in this letter too.

·

Basic reporting

This paper presents the improved related-key differential neural distinguishers for Simon32/64 by using other input differences. Additionally, by employing two input differences, more accurate RKNDs are provided.

Experimental design

no comments

Validity of the findings

1. It should be noted that the paper [1] has already proposed the method of using two input differences. Therefore, the author's contribution in this regard may be an invalid contribution.

2. As Reviewer 2 commented: "The use of two different input differences provides more information in the ciphertext, which naturally results in an improved distinguisher."

3. Regarding the feasibility of a successful key recovery attack, the authors have responded that there are experiments where the complexity has been reduced. But they did not give such examples in their responses or in their paper.

[1] WANG, Gao, Gaoli WANG, and Siwei SUN. "Investigating and Enhancing the Neural Distinguisher for Differential Cryptanalysis." IEICE TRANSACTIONS on Information and Systems 107.8 (2024): 1016-1028.

Additional comments

1. Please explain the difference between this paper's contribution and that of the paper [1].

2. To demonstrate that the distinguishing task in this paper indeed improves upon that of Lu et al., please provide the accuracy of the distinguisher in the paper when applied to the dataset generated in the task model of Lu et al., where the negative samples are generated from random numbers. If the accuracy is higher than that of Lu et al., it indicates an improvement; otherwise, as Reviewer 2 mentioned: “- In Formula 10, the design of positive and negative samples differs from previous differential neural distinguishers, with the negative samples possibly being non-random. Therefore, it is inappropriate to compare the results of this paper with those of Lu et al., as they do not address the same task.”

3. Please provide the example of successful key recovery mentioned by the authors in their response.

[1] WANG, Gao, Gaoli WANG, and Siwei SUN. "Investigating and Enhancing the Neural Distinguisher for Differential Cryptanalysis." IEICE TRANSACTIONS on Information and Systems 107.8 (2024): 1016-1028.

Reviewer 2 ·

Basic reporting

no comments

Experimental design

no comments

Validity of the findings

no comments

Additional comments

all my questions have been well addressed. I this this paper can be accepted as it is.

Reviewer 3 ·

Basic reporting

The authors have well answered my questions. I have no question now.

Experimental design

good and detailed.

Validity of the findings

novel enough for publication.

---

## Round 0.3 · Minor Revisions

According to the reviewer's comments, your paper is ready for publication. However, when I deeply checked your paper, I found some weaknesses. I think that when these additions are made, the quality of your paper will increase. As an annotated PDF (100801-v2-required additions.pdf), ı will upload your manuscript that contains my additional comments. I highlighted some parts and added the comments. Please make some generalizations if the required changes match the paper's other parts. Please prepare a response letter and add the necessary changes to this letter. Answering the queries informally in the response letter will not be enough, so fulfills the required changes to your manuscript.

·

Basic reporting

no comments

Experimental design

no comments

Validity of the findings

no comments

Additional comments

The authors have addressed my inquiries, and I currently have no further questions.

Reviewer 2 ·

Basic reporting

I had accepted this paper in the previous round review.

Experimental design

Good

Validity of the findings

It is interesting

Reviewer 3 ·

Basic reporting

no comment

Experimental design

no comment

Validity of the findings

no comment

Additional comments

The authors have well answered my questions, I have no question now.

---

## Round 0.4 · accepted · Accept

The final version of your manuscript has met the reviewer's comments and my suggestions. Since I gave the last suggestions, I did not send the answers again to the reviewers and ı think this version is ready for publication. Please recheck your manuscript and ensure there are no grammatical or typos to decrease the quality of your paper in the publication process.